# Study on Clamping Mechanism of Internal and External Variable Diameter Lifting Tool for Offshore Foundation Pile

**Zhuo Wang** [1,*], **Zhuang Li** [1], **Tao Wang** [2] **and Bo Zhang** [1]

1   College of Mechanical and Electrical Engineering, Harbin Engineering University, Harbin 150001, China; lz942178174@hrbeu.edu.cn (Z.L.); zhangbo_heu@hrbeu.edu.cn (B.Z.)
2   School of Mechanical Engineering, Hebei University of Technology, Tianjin 300130, China; 18846166436@hrbeu.edu.cn
*   Correspondence: wangzhuo_heu@hrbeu.edu.cn

**Abstract:** Large marine foundation piles are an important part of offshore structural pile foundations, and their lifting operations have always been a major problem in the construction and construction of marine structures. Based on IHC's bilateral marine foundation pile spreader, this paper proposes a structural scheme of "internal and external clamping type variable diameter marine foundation pile spreader". It solves the problem of poor adaptability of spreaders to foundation piles of the same specification and different pipe diameters. At the same time, this article has conducted in-depth research on the two clamping methods of friction clamping and wedge tooth embedded clamping. Through experiments, it is found that under the same lateral load, the load capacity of the wedge teeth tightening is three times that of the friction clamping. Aiming at the embedding and clamping method of the wedge teeth of the spreader, first of all, the influence of the tooth profile angle of the wedge teeth on their embedding performance was studied by the plastic mechanics slip line field theory and Abaqus simulation analysis. Subsequently, the elastic mechanics theory and Abaqus simulation analysis were used to study the stress characteristics of the wedge teeth during the lifting process, and the internal stress distribution was obtained. The article aims to provide a reference for the design of spreaders in actual projects.

**Keywords:** marine foundation pile; large-scale spreader; clamping and load-bearing mechanism; wedge embedding performance; plastic deformation simulation

## 1. Introduction

In offshore waters, pile foundations are a common form of foundation for offshore structures. Steel piles driven into the seabed as supporting members of pile foundation are often referred to as offshore foundation piles. Compared with land steel piles, offshore piles have the structural characteristics of long length, large diameter, large weight and high strength [1]. In order to ensure the overall strength of the foundation pile and maintain good mechanical properties and stability during piling, avoid stress concentration and buckling deformation, the outer surface of the foundation pile is always designed as a smooth cylindrical surface, meaning that there is often no ideal lifting point on the surface, which also brings technical difficulties to the lifting and handling of the foundation pile [2]. In order to solve the lifting problem of foundation piles, marine engineers at home and abroad have developed and designed many special lifting tools. The common types of foundation pile lifting tools are unilateral, bilateral, external and internal expansion lifting tools [3].

At present, the design and production of large-scale marine foundation pile clamp and spreader are mainly monopolized by the IHC Company of The Netherlands [4]. IHC Company produces a variety of spreaders, including saddle & hook type pile spreader [5], bilateral clamping type pile spreader [6], external lifting tool type pile spreader [7], and

international lifting tool type pile spreader [8]. These spreaders can complete the lifting operation of up to 1000 t foundation piles.

For the two clamping modes of hanger clamping mechanism, friction clamping is to produce friction by extruding the pipe wall of foundation pile, thus realizing the clamping function. On the basis of friction clamping, wedge teeth are embedded in the original clamping parts to press wedge teeth into the pipe wall of foundation pile to achieve clamping. Therefore, to study the deformation of foundation pile under the action of lifting device clamping mechanism is essentially to study the deformation of foundation pile under the action of external load squeezing. In view of the structural characteristics of large length-diameter ratio and thin wall of pile, it can be regarded as an elastic thin-walled cylindrical shell structure for mechanical solution. There are many related studies on this kind of structure at home and abroad. This paper mainly studies the force and deformation of foundation pile under the action of hanger clamping, which belongs to the field of static solution of thin shell structure.

In view of the structural characteristics of large length-diameter ratio and thin wall of pile, it can be regarded as an elastic thin-walled cylindrical shell structure for mechanical solution. There are many related studies on this kind of structure at home and abroad. This paper mainly studies the force and deformation of foundation pile under the action of hanger clamping, which belongs to the field of static solution of thin shell structure.

In 2012, Brischetto and Carrera of Turin Polytechnic University (Turin, Italy), proposed an improved finite element shell model with eight nodes, each node having nine degrees of freedom [9]. In 2016, Zheng from Zhejiang University (Hangzhou, China) used the vector finite element method to analyze the complex mechanical behavior of thin metal shells, and gave an application example of the vector finite element method [10]. In 2016, Tullu and Kang at Pusan National University (Pusan, Korea) studied the elastic deformation of fiber reinforced composite stiffness cylindrical shells, and gave numerical calculation examples for solving the stress and deformation of such cylindrical shells [11].

When inserting and clamping the foundation pile with the wedge teeth, the interaction between the wedge teeth and the wall of the pile pipe is complicated. During the clamping process, the pile wall undergoes the process of elastic deformation to plastic failure, that is, the transformation from small deformation (linear process) to large deformation (non-linear process).Many scholars have studied the finite element simulation analysis of the plastic deformation and failure process of metals, and the theoretical methods involved are mainly rigid-plastic nonlinear finite element methods.

In 2010, Jayasekara and Hwang of Renhe University (Incheon, Korea) made a finite element analysis of the riveting process of sheet metal and equationted a criterion for evaluating the riveting strength [12]. In 2016, Yan, Hua and Zhao of Taiyuan University of Science and Technology (Taiyuan, China) used least square meshless method to simulate the plastic deformation process of metals [13]. In 2018, Jackson performed a finite element simulation study on the process of indentation on the extruded cylinder surface, obtained the general law of surface deformation of the cylinder [14].

Through a comparative analysis of the structural characteristics and clamping methods of various existing large-scale marine foundation pile spreaders, it is found that existing foundation pile spreaders often clamp the pipe wall of the foundation pile on one side, and the foundation pile is prone to large It is deformed by force, and the adaptability of the spreader to the foundation piles of the same specification and different pipe diameters is poor. Therefore, it is necessary to study the optimization of spreader structure. At the same time, as the core structure of the spreader, the clamping mechanism is of great significance for the in-depth study of its clamping method. In this paper, IHC's bilateral marine foundation pile spreader is used as a blueprint, and the structure of the spreader is optimized so that it can adapt to different diameters of the foundation pile. At the same time, through experimental research, theoretical analysis and simulation analysis, in-depth study of the two clamping methods of friction clamping and wedge tooth clamping.

Based on the comparative analysis of the structures and clamping modes of various kinds of marine foundation pile spreaders at present, this paper puts forward a design scheme of internal and external clamping type marine pile spreader with changeable diameter, aiming at the problems of poor stress and deformation of the marine piles while the existing foundation pile spreaders under clamping conditions and poor adaptability of spreaders to marine piles of the same specification with different diameters. The spreader designed in this paper can be compatible with friction clamping and wedge tooth insertion clamping. It can complete the lifting work of series of foundation piles with shell thickness of 25 mm, length of 50 m, inner diameter of 4.0 to 4.5 m and outer diameter of 4.05 to 4.55 m. It improves the versatility of the use of marine foundation pile spreaders and reduces the operation cost. Through theoretical calculation and simulation analysis, the internal and external clamping method proposed in this paper can effectively alleviate the stress and deformation of foundation piles and ensure their contour under the premise of guaranteeing the reliable clamping of spreaders, which provides a reference for the practical application of engineering.

## 2. Experimental Method on the Clamping Mechanism of Spreaders

### 2.1. Load-Bearing Experiment of Friction Clamping of Spreader

According to the friction clamping mode of the hanger, relevant experiments are designed. The specific experimental process is shown in Figure 1. In the course of the experiment, the jack is used to simulate the clamping cylinder of the hanger in the horizontal direction, and the lateral clamping load is applied to the pile specimens. The force sensor detects the corresponding loading force. In the vertical direction, jack is used to simulate the self-weight of pile, and force sensor is used to detect the corresponding load value. In the process of longitudinal loading, when the displacement of the pile specimen changes obviously, the friction clamping failure occurs at this time. The data of the longitudinal force sensor recorded at this time are shown in Table 1.

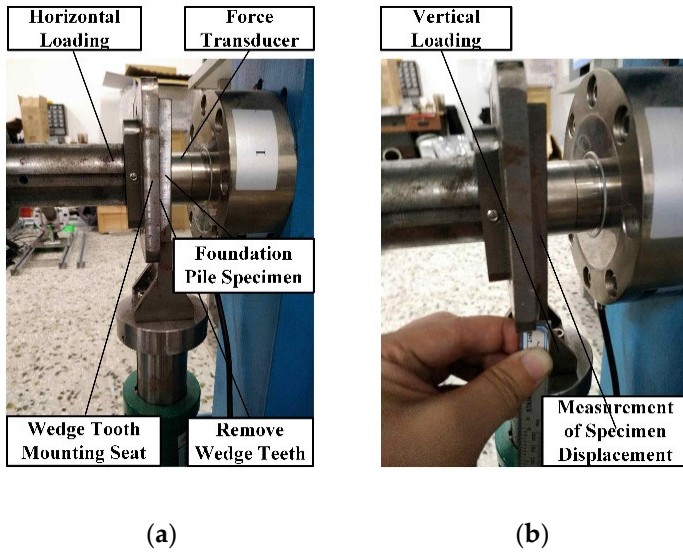

(**a**)                                              (**b**)

**Figure 1.** Experimental Process of Friction Clamping Load of Spreader. (**a**) Horizontal load chart, (**b**) Vertical load chart.

**Table 1.** Data of friction clamping of a spreader.

| Horizontal Loading Force, [$10^4$ N] | 4.9 | 9.8 | 1.5 | 2.0 | 2.5 | 3.0 |
|---|---|---|---|---|---|---|
| Maximum Longitudinal Loading Force, [$10^4$ N] | 2.1 | 4.3 | 6.5 | 8.7 | 1.1 | 1.3 |

Based on the data in Table 1, the maximum static friction coefficient $\mu_{max}$ between the pile specimen and the experimental clamping mechanism can be obtained by Equation (1):

$$\mu_{\max} = \frac{(F_v/2)}{F_h} \tag{1}$$

here, $F_v$ is the longitudinal loading force ($10^4$ N) and $F_h$ is the lateral loading force ($10^4$ N).

According to the experimental data, the maximum static friction coefficient of each group of experiments is calculated, and the relationship between the static friction coefficient and the horizontal loading force is drawn as shown in Figure 2.

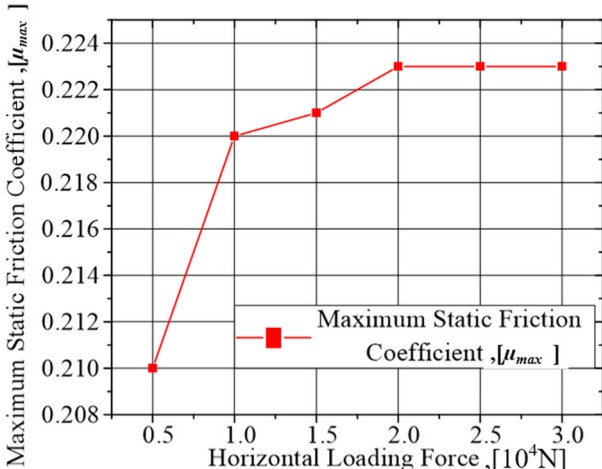

**Figure 2.** The relationship between the maximum static friction coefficient of micron and the horizontal loading force.

From Figure 2, it can be seen that the maximum static friction coefficient between the contact surfaces of the experimental device increases with the increase of the horizontal loading force at the initial stage, and then stabilizes at 0.223, so the maximum static friction coefficient of the friction clamping mode of the hanger is 0.223.

### 2.2. Load-Bearing Experiment of Wedge-Shaped Teeth Embedded in Suspension

In order to study the influence of the tooth profile angle of the wedge block on its embedding performance, single tooth embedding experiments were carried out on four kinds of single tooth wedge teeth with tooth profile angles of 45°, 60°, 75° and 90°. Lateral load is gradually applied through the horizontal jack, and the wedge teeth are pressed into the test piece after being loaded. At the same time, the load cell measures the load force in real time and transmits it to the upper computer for display and recording. During the loading process, the loading force is controlled according to the real-time value displayed by the host computer until the target load is reached. After each loading, use a digital cursor caliper with an accuracy of 0.01 to measure the embedding depth of the wedge teeth after loading and record. The relationship between the embedded depth of single-tooth wedge blocks with different tooth angles and the loading force is shown in Table 2.

**Table 2.** The relationship between the embedding depth and loading force of single tooth wedges with different tooth angles.

| Lateral load, [$10^4$ N] | 0.5 | 1.0 | 1.5 | 2.0 | 2.5 | 3.0 |
|---|---|---|---|---|---|---|
| **45 °single-tooth wedge's embedding depth, [mm]** | 0.02 | 0.05 | 0.07 | 0.09 | 0.12 | 0.15 |
| **60 ° single-tooth wedge's embedding depth, [mm]** | 0.02 | 0.04 | 0.06 | 0.07 | 0.10 | 0.12 |
| **75 °single-tooth wedge's embedding depth, [mm]** | 0.02 | 0.03 | 0.05 | 0.07 | 0.09 | 0.11 |
| **90 °single-tooth wedge's embedding depth, [mm]** | 0.01 | 0.02 | 0.04 | 0.06 | 0.08 | 0.10 |

Observing the experimental results, under the same loading force, as the tooth profile angle increases, the embedding depth of the wedge teeth decreases. This shows that for the clamping method of the wedge teeth of the spreader, in order to improve the clamping effect, on the premise of ensuring the strength of the wedge teeth, the wedge teeth with a smaller tooth angle should be selected as much as possible to ensure the wedge teeth Pointed sharpness. The plastic deformation caused by lifting tools to the internal part of the pile is caused by a small area, which does not damage the overall strength of the pile.

In order to study the clamping bearing capacity of wedge teeth embedded in suspension, several specimens of wedge teeth and foundation piles with different specifications are designed and manufactured.

On the basis of Figure 1, wedge teeth and pile specimens are installed in the experimental device, and the clamping load experiment of wedge teeth embedded in the hanger is carried out. In order to study the stress and deformation of wedge teeth in the process of loading, the resistance strain gauge is pasted at the designated position of wedge teeth and pile specimens. The specific pasting method is shown in Figure 3.

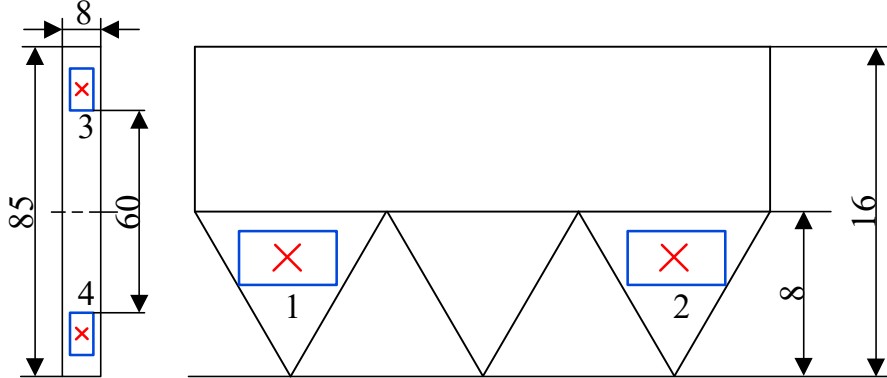

**Figure 3.** Insertion of Teeth of Spreader to Clamp Experimental Strain Gauge.

In the course of the experiment, wedge teeth are firstly clamped by a horizontal jack to embed the wedge teeth in the pile specimens, and then the vertical jack is applied to simulate the self-weight of the pile. Use digital cursor calipers to observe and measure the displacement of the test piece. At the same time, use strain gauges attached to the wedge teeth and the test piece to detect the force and deformation of each part. The vertical load cell measures the loading force in real time. During the loading process, when there is a significant displacement of the test piece or a significant change in the reading of the cursor caliper, it means that the clamping mechanism has failed to clamp, and the value of the longitudinal loading force at this time is the ultimate bearing capacity of the clamping mechanism. In the process of longitudinal loading, when the load is about 4.0 T, the specimen of foundation pile loosens and the displacement changes obviously,

which indicates that the wedge tooth insertion clamping failure at this time. Although the contact surface between wedge teeth and specimens is grooved after embedding in pile specimens, referring to the definition of equivalent friction coefficient in thread pair friction calculation, the equivalent friction coefficient between wedge teeth and specimens is calculated by substituting transverse load 3.0 T and longitudinal load 4.0 T Equation (1), and the equivalent friction coefficient $F_V$ between wedge teeth and specimens is 0.667.

The experimental results show that under 3.0 T lateral clamping load, the maximum static friction coefficient of the friction clamping mode is 0.223, while the equivalent friction coefficient of the wedge insertion clamping mode is 0.667, which is approximately three times the relationship between them. This shows that although wedge teeth are embedded in a small depth, they have a strong bearing capacity. The reliability of wedge teeth embedded clamping mode is verified, and the carrying capacity of the two clamping modes of hanger is quantitatively compared and analyzed.

According to Figure 4, strain gauges 1, 2 and 3, 4 reflect the stress and deformation of wedge tooth tip under transverse load and the stress and deformation of specimens, respectively. During the longitudinal loading process, through uniaxial strain gauges measure the strain of different measuring points, the strain values of each measuring point are shown in Table 3.

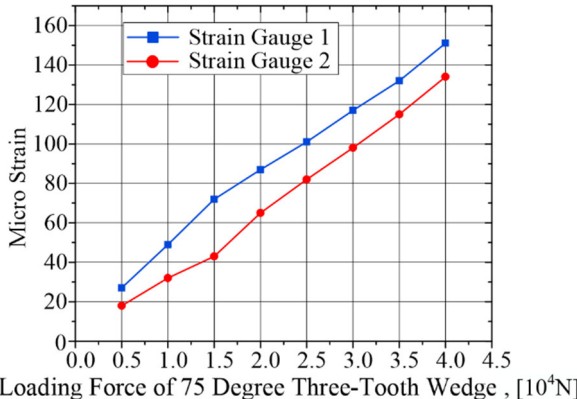

**Figure 4.** Deformation of Wedge Tooth under Longitudinal Load.

**Table 3.** The strain values of each measuring point are under different loading force of 75° three-tooth wedge teeth.

| Micro Strain / Measuring Point — Loading Force, [$10^4$ N] | Strain Gauge 1 | Strain Gauge 2 | Strain Gauge 3 | Strain Gauge 4 |
|---|---|---|---|---|
| 0.5 | −27 | −18 | −25 | −13 |
| 1.0 | −49 | −32 | −39 | −22 |
| 1.5 | −72 | −43 | −48 | −36 |
| 2.0 | −87 | −65 | −56 | −41 |
| 2.5 | −101 | −82 | −65 | −53 |
| 3.0 | −117 | −98 | −77 | −65 |
| 3.5 | −132 | −115 | −92 | −78 |
| 4.0 | −151 | −134 | −105 | −92 |

In order to intuitively reflect the variation of strain values of each measuring point with the vertical loading force, according to the data in the table, Figure 4 is made.

From Table 2 and Figure 4, it can be seen that the wedge teeth are bended as a whole under longitudinal loading, and the strain gauge can detect the degree of longitudinal compression. The stress and deformation of the wedge teeth are not uniform. The closer the wedge teeth are to the action area of external load, the greater the deformation degree of the wedge teeth.

### 3. Analysis of the Bearing Mechanism of Friction of Lifting Tools

#### 3.1. Study on Safe Lifting Conditions for Friction Clamping of Lifting Tools

The principle of lifting the foundation pile by friction clamp is relatively simple. Considering the actual lifting process, the crane cable is connected to the lifting point of the spreader, and a vertical upward lifting force $R$ is applied. The pulling force of the lifting cable is transmitted to the foundation pile through the spreader clamping mechanism, thereby realizing the lifting of the foundation pile.

Bear in mind that the friction between the four external clamping wedges of the hanger and the outer wall of the pile is $f_1, f_2, f_3, f_4$, the friction between the internal supporting parts and the inner wall of the pile is $f_1, f_2, f_3, f_4$, the self-weight of the hanger is $G_1$, and the self-weight of the pile is $G_2$.

For the friction clamping mode of hangers, the lifting of foundation piles should satisfy the following three conditions:

(1)   Before lifting, the lifting device should be in stable position on the foundation pile, that is, the friction clamping force of the lifting device should be able to overcome the influence of the gravity of the lifting device itself, and ensure that the lifting pulling force $R$ is removed after the lifting device is in place, so that the lifting device will not overturn.

(2)   In the lifting process, the friction clamping force provided by the hanger should meet the lifting requirements and have a certain safety factor.

(3)   Because there are many uncertainties in the lifting process, and the tension of the crane cable may change, the clamping mechanism of the hanger should be able to withstand certain impact vibration and sudden change of load.

The three-dimensional space force system is composed of lifting tension $R$, lifting gear gravity $G_1$, pile gravity $G_2$ and friction clamping force. It is difficult to determine the direction of static friction force here. Because the contact surface is circular, the direction of static friction must be tangent to the contact surface. Combining with the specific structure size of the hanger, after the hanger is clamped at the top of the pile, the direction of friction clamping force of the clamping mechanism of the hanger is shown in Figure 5. For the first lifting condition, and assuming that the lifting gear is installed horizontally, the force acting on the lifting system is shown in Figure 6.

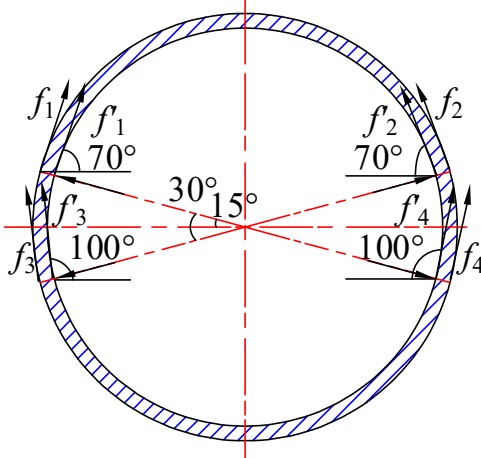

**Figure 5.** Direction and Action Point of Friction Clamping Force of Spreader.

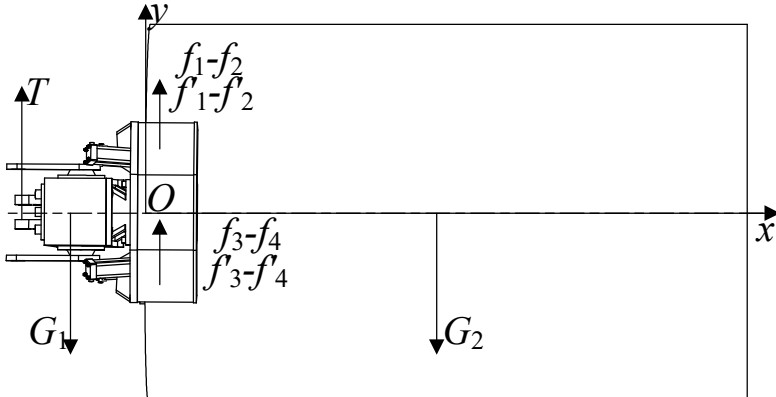

**Figure 6.** Force Condition during Installation and Positioning of Spreader.

Although the force system in Figure 6 is a space force system, in the graphical coordinate system, according to the balance relationship in the Y direction of the force system, the suspender will not overturn when it is in stable position, and the friction clamping force of the suspender should not exceed the maximum static friction force. Regarding the solution of the maximum static friction coefficient between the spreader clamping mechanism and the contact surface of the foundation pile, it is a method of solving the maximum static friction coefficient based on statistical principles proposed by You Jinmin of Xi'an Jiaotong University [15].

Although the force system in Figure 6 is a space force system, under the coordinate system shown in the figure, according to the equilibrium relationship of the y-direction of the force system, the spreader is stable in position without overturning, to satisfy the following Equation (2), and each friction clamping force cannot exceed the maximum static friction:

$$G_1 = (\sum_{i=1}^{4} f_i \cos\theta_1 + \sum_{j=1}^{4} f_j' \cos\theta_2) \tag{2}$$

For the second lifting condition, the force relationship of the lifting system is shown in Figure 7.

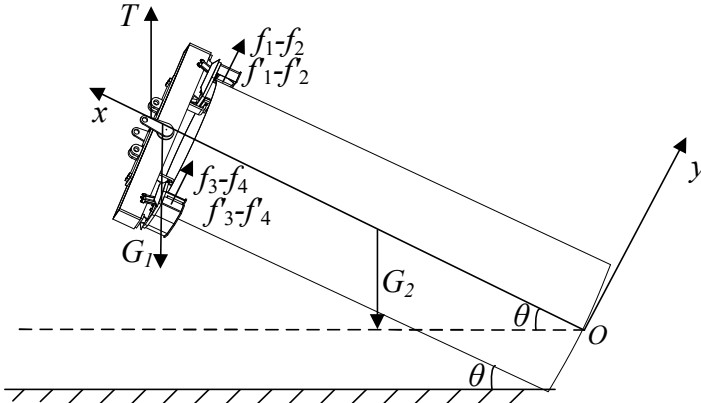

**Figure 7.** Stress Condition of Foundation Pile during Lifting Process.

In order to ensure the stability of the lifting process, at each moment, corresponding to each $\theta$ angle, the following Equation (3) is to be satisfied:

$$T\cos\theta + (\sum_{i=1}^{4} f_i \cos\theta_1 + \sum_{j=1}^{4} f_j' \cos\theta_2) = (G_1 + G_2)\cos\theta \tag{3}$$

In order to ensure the stability of the lifting process, the Equation (2) should be satisfied for every $\theta$ angle at each moment. At the same time, each friction clamping force shall not exceed the maximum static friction force. To sum up, Equations (2) and (3) can be used as checking equations for safety lifting conditions of suspension when friction clamping method is adopted, and the safety of lifting process can be judged and analyzed.

First of all, for the first condition, the weight of the spreader is about 20 T, and the maximum static friction force of each clamping mechanism is 24.41 kN, where $\theta_1 = \theta_2 = 20°$, substituting Formula (3), it is found that the clamping provided by the spreader. The force cannot meet the weight of the spreader. The spreader cannot be installed on the wall of the pile pipe and will overturn. If the thrust of the hydraulic cylinder is increased, the frictional clamping force will also increase. The self-locking feature can be realized by using the friction clamping method to complete the lifting of the foundation pile. Since the spreader developed in this paper needs to take into account the two clamping methods of friction clamping and wedge tooth embedded clamping, the final parameters of the hydraulic cylinder should be considered comprehensively in combination with the two working conditions.

When the sling is clamped by friction, the clamping force of the sling is different according to the weight of the pile. For small-tonnage foundation piles, the clamping force of hangers is small, and the deformation of foundation piles is within the elastic range.

The preliminary parameters of the clamping cylinder are shown in Table 4.

**Table 4.** Structural and Technical Parameters of Clamping Hydraulic Cylinder.

| Work Press, [MPa] | Allow Moving Distance, [mm] | Piston Dia, [mm] | Piston Rod Dia, [mm] | Thrust, [kN] | Tension, [kN] |
|---|---|---|---|---|---|
| 30 | 330 | 120 | 65 | 339.3 | 239.7 |

In the research and design process of the suspension described in this paper, the design index of the suspension is shown in Table 5.

**Table 5.** Technical Requirements for the Internal and External Clamping Type Marine Pile Spreader.

| Design Items | Technical Requirements |
|---|---|
| Target Thickness of Marine Foundation Piles | 25 mm |
| External Diameter Range of Target Marine Foundation Piles | 4.05 to 4.55 m |
| Internal Diameter Range of Target Marine Foundation Piles | 4.0 to 4.5 m |
| Weight Range of Target Marine Foundation Piles | 126,400 to 142,100 kg |
| Target Length of Offshore Foundation Piles | 50 m |
| Working Pressure of Lifting Hydraulic System | 30 MPa |
| Variable Diameter Speed of The Spreader | 1 mm/s |

*3.2. Analysis of Load-Bearing Mechanism of Wedge-Shaped Teeth Embedded Clamping of Spreader*

When the suspension is clamped with wedge teeth, the clamping mechanism works and the wedge teeth will squeeze the outer wall of the foundation pile and eventually pierce the surface of the foundation pile, resulting in the plastic deformation of the foundation pile, which is embedded in the outer wall of the foundation pile. In this way, when the hanger completes the clamping action and lifts, under the action of the gravity of the pile itself, the squeezing force and friction force formed between the wedge teeth and the pipe wall of the pile become a new clamping force. The interaction between wedge teeth of clamping mechanism and pipe wall of foundation pile is shown in Figure 8.

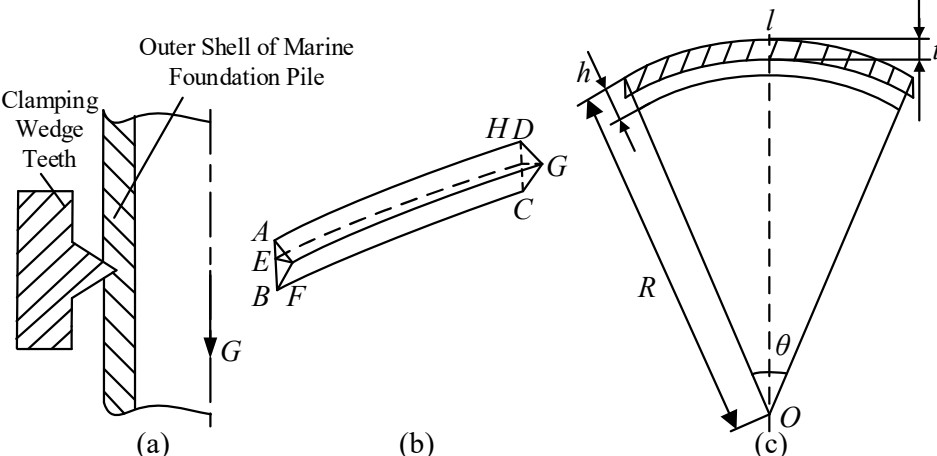

**Figure 8.** Interaction between Wedge Teeth of Clamping Mechanism and Pipe Wall of Foundation Pile. (**a**) Force exerted by single clamping teeth. (**b**) Single clamping tooth shape unit, (**c**) the geometric relationship between the wedge teeth and the foundation pile.

In Figure 8a, single tooth *ABCDFG* is taken as the research object, and its tooth shape angle is $\varphi$, as shown in Figure 8b. Taking *EFGH* as its normal middle surface, the normal depth of wedge teeth embedded in the outer wall of pile is *t*. The radius of the bus bar of the wedge teeth is the same as the outer diameter *R* of the foundation pile, and the arc length *l* is a known quantity. At this time, the geometric relationship between the wedge teeth and the foundation pile is shown in Figure 8c, where *h* is the known thickness of the foundation pile wall and $\theta$ is the central angle of the wedge teeth to the center of the foundation pile.

According to the geometric relationship in the figure, and omitting the small geometric quantity, the area $A_1$ of the wedge teeth embedded in the outer wall of the foundation pile is calculated, and then the contact area $A_2$ of the wedge teeth working face and the foundation pile is calculated by using the tooth angle $\varphi$ Equation (4):

$$\begin{cases} A_1 = -\frac{1}{2}t^2\theta + lt \\ A_2 = \frac{1}{\cos(\frac{\varphi}{2})}(-\frac{1}{2}t^2\theta + lt) \end{cases} \tag{4}$$

In practical engineering, the clamping force *F* produced by wedge tooth insertion is related to lifting force and pile self-weight, but according to the geometrical relationship between wedge tooth and pile pipe wall, it can be simply regarded as proportional to $A_1$ and $A_2$, that is, the larger the contact area between wedge tooth and pile, the greater the clamping force produced. In the expression, because the geometrical size of wedge teeth has been determined, the arc length *l* and the center angle $\theta$ have been determined, so the key factors affecting the clamping force are the insertion depth *t* of wedge teeth and the wedge teeth shape angle $\varphi$.

### 3.3. Slip Line Field Method for Solving Plastic Deformation Process of Foundation Piles

Because of the large curvature radius of the outer wall of the pile, the process of inserting the wedge teeth of the hanger clamping mechanism into the pipe wall of the pile can be simplified as the process of pressing the plastic base material with a straight rigid wedge as shown in Figure 9. This kind of problem belongs to "unsteady plastic flow" problem, which is difficult to solve directly. Prague has done some research on this kind of problem, which transforms the problem into a quasi-steady flow problem by using geometric assumptions, so that the slip line field theory can be used to solve [16].

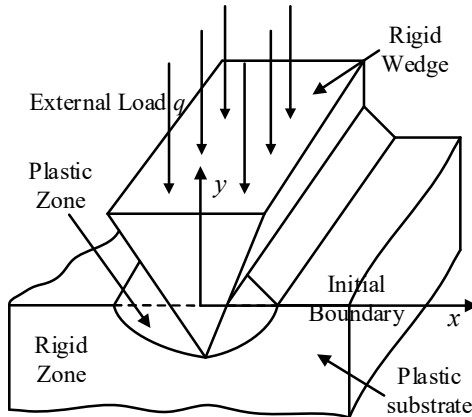

**Figure 9.** Pressure Model of Rigid Wedge.

For such problems, according to Prague and Hodge's research ideas, the slip line field in the process of rigid wedge embedding is shown in Figure 9. Because of the symmetry of the problem, half of the structure is taken to study. In Figure 9, *BH* is the free surface of the plastic body and △*ABG* is the protruding part of the plastic body extruded by the wedge from the base plane.

The length of the *AB* edge is *a*, the vertical distance from the center point *O* to the *AB* edge is *b*, and the distance from the *A* point to the reference plane is *h*. Under the external load *q*, the embedded depth of the wedge is *t* after 1 of movement per unit velocity, and the tooth angle of the wedge is 2*β*. △*ACD* is a centrifugal sector surrounded by a group of slip lines, and its central angle is *α*. Because *AC* and *AD* are *α* lines of slip line field, *BC* and *DE* are *β* lines, then *AC*⊥*BC*, *AD*⊥*DE*.

According to the geometric relationship in Figure 10, the relationship between the wedge insertion depth *t* and the external load *q* and the wedge tooth angle *β* is obtained as follows.

$$q = 4\tau_s t(1+\alpha)\frac{\sin\beta}{\cos\beta - \sin(\beta - \alpha)} \tag{5}$$

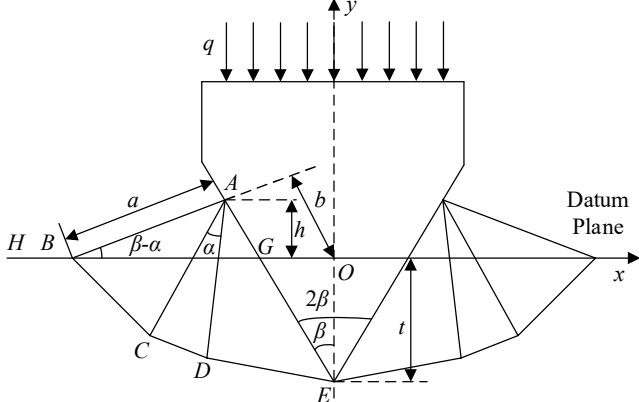

**Figure 10.** Slip Line Field of Rigid Wedge Pressing into Plastic Body.

According to Equation (5), the relationship between wedge insertion depth and wedge tooth angle and external load is given as shown in Figure 11.

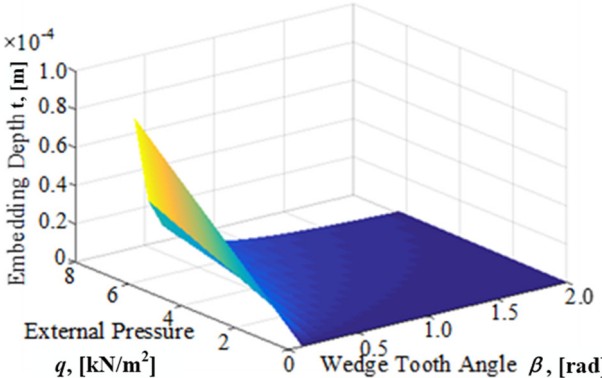

**Figure 11.** The Relationship between Wedge Insertion Depth and External Load and Wedge Tooth Angle.

From Figure 11, the wedge embedding depth $t$ increases linearly with the increase of positive pressure $q$ of external load, while the smaller the wedge tooth angle $\beta$ and the sharper the wedge, the greater its embedding depth. In general, it is necessary to apply a large force load to the rigid wedge, so that the wedge can move down by a small amount of displacement. However, because the output of the hydraulic cylinder is constant, the tooth profile angle of the wedge determines the depth of insertion of the wedge. The key factors for the reasonable selection of the wedge tooth profile angle are of great significance in practical engineering.

### 3.4. Simulation Analysis of Plastic Embedding of Single-Row Wedge Teeth

In this section, the wedges with 45°, 60°, 75° and 90° are selected to simulate the wedge teeth, and the finite element simulation analysis of the embedding process is carried out. Because of the large radius of curvature of the outer diameter of the pile, it can be equivalent to a straight line in a certain range. In addition, the volume of the model should be increased appropriately in the simulation process, so that the stress diffusion in the embedding process can be obtained more clearly.

In this paper, a three-dimensional simulation is made. In addition to guaranteeing the angle, the wedge has a tooth height of 8 mm, a total height of 16 mm, a total length of 85 mm and a cube size of 85 × 85 × 8 mm.

Because Abaqus plastic deformation simulation needs real strain parameters of material, but because the material of foundation pile is super-austenitic stainless steel (254SMo) or other corrosion-resistant alloy steel, its real strain data is difficult to obtain. This paper is limited by the research conditions, and does not have the conditions for material mechanics experiment, so SA283D material which has been strengthened is selected as the foundation pile specimens. These two materials are plastic materials. The plastic yield process of these two materials is the same, only the yield point values are different. The general rule of plastic deformation of pile in wedge tooth embedding process can be obtained by using this alternative method [17].

Generally, X45NiCrMo4 cold working die steel is used as the material of wedge teeth of clamping mechanism, its tensile strength can reach 1800 MPa, and the hardness of surface treated teeth can reach HRC60 [18]. But due to the lack of real stress and strain parameters of such materials in the simulation process, In the following, we will define the wedge as a pure ideal rigidity and an elastic body with a sufficiently large elastic modulus for simulation analysis. Although it is different from the actual result, it can reflect the general law. In practical engineering, the piston rod of hydraulic cylinder extends uniformly. In order to simulate the actual working condition, the specimen is fixed in the simulation process, and the wedge is given downward displacement load to make it move downward uniformly. The process of embedding the specimen is observed. The wedge speed selected in this paper is 1 mm/s.

The mechanical properties and real stress-strain parameters of pile specimens during the simulation process are shown in Tables 6 and 7.

**Table 6.** Mechanical properties parameters of SA283D (Strengthened).

| Material | Density, [g/cm$^3$] | Modulus of Elasticity, [GPa] | Yield Strength, [MPa] | Poisson Ratio |
|---|---|---|---|---|
| SA283D | 7.85 | 210 | 418 | 0.3 |

**Table 7.** SA283D (Strengthened) Real Stress-Strain Parameters.

| Real Stress, [MPa] | 418 | 500 | 605 | 780 | 921 |
|---|---|---|---|---|---|
| True Strain | 0.00783 | 0.01581 | 0.02983 | 0.095 | 0.45 |

In order to further understand the plastic deformation process of the foundation pile, in this section, the finite element method will be used to simulate the process of embedding the wedge teeth into the foundation pile. Since the contact and plastic deformation problems are more nonlinear, the simulation environment is Abaqus software. In actual engineering, the piston rod of the hydraulic cylinder is extended at a constant speed. In order to simulate the actual working conditions, the test piece is fixed during the simulation process, and the wedge is loaded with a downward displacement to make it move downward at a uniform speed. Process, the wedge speed selected in this article is 1 mm/s. In terms of meshing and cell selection. Although the plastic deformation simulation conducted in this paper has a small amount of deformation, no element reconstruction is required. However, it is still much larger than the elastic deformation of the specimen, and a small amount of mesh distortion and distortion may appear during the iteration process. Therefore, the 8-node linear hexahedral element C3D8R is selected, and the reduced integration and hourglass control are performed to prevent the problem of shear blocking during the simulation.

After simulation calculations, the four wedge blocks with tooth angles of 45 degrees, 60 degrees, 75 degrees and 90 degrees were assigned displacements during walking and after embedding. Take the cross-section observation, the Mises stress distribution of the specimen is shown in Figure 12.

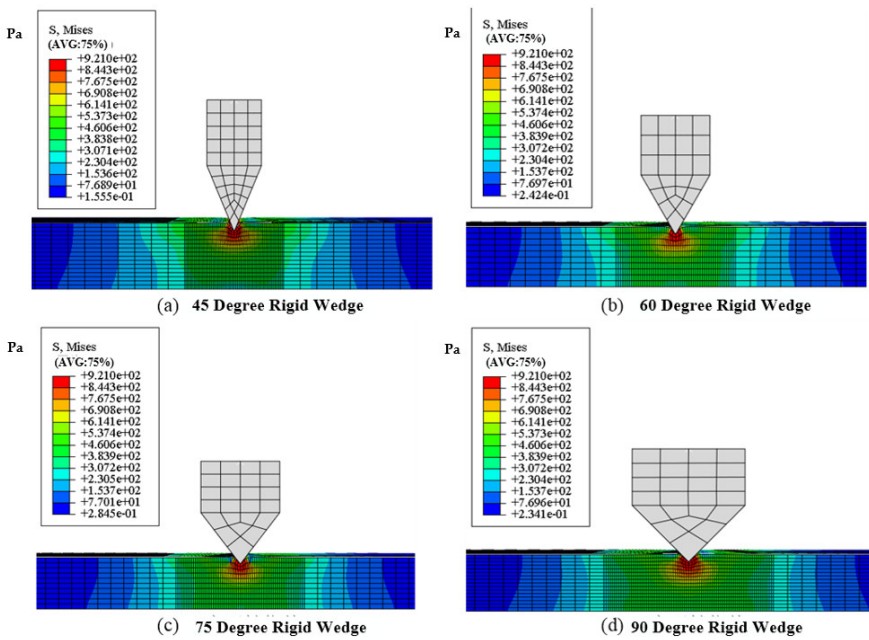

**Figure 12.** Stress Distribution of Specimens Embedded with Rigid Wedges of 45°, 60°, 75° and 90°.

The simulation results show that the Misses stress distribution in the pile is similar when four kinds of rigid wedges with tooth angles are embedded in the pile specimens. The maximum stress occurs in several elements directly contacted with the wedge teeth. By observing the whole embedding process, the stress of several elements in direct contact with the wedge varies sharply. After reaching the yield stress, the plastic zone expands to a certain extent, and the wedge can continue to move downward. The extension form of the plastic zone is similar to the shape of the stress field obtained by using the slip line field theory.

Because the displacement of the wedge increases uniformly with time and the specimen is fixed, the reaction force of the specimen to the wedge is the concentrated external load applied to the center of the wedge surface. The relationship between the downward displacement of the wedge and the external load applied on the wedge is shown in Figure 13.

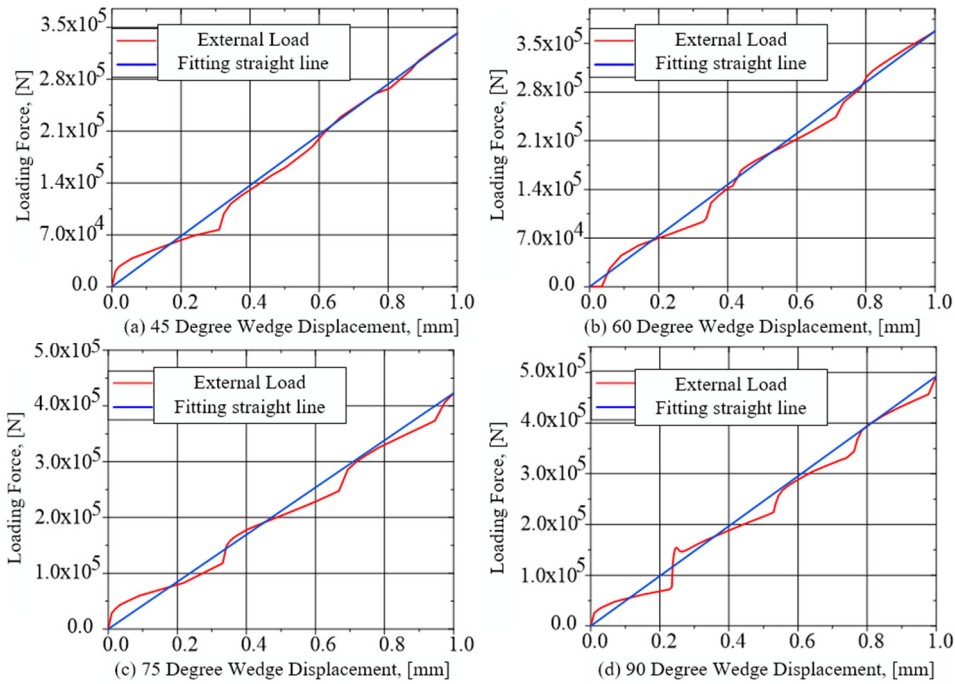

**Figure 13.** Change of Loading Force and Displacement during Wedge Embedding with 45°, 60°, 75° and 90° Tooth Angle.

From Figure 13, it can be seen that in order to make the wedge move downward uniformly, the change of the external load imposed on the wedge with the displacement is not linear. The reason is that the specimen undergoes several stages of elastic deformation, plastic yield and plastic flow, and the loading force corresponds to these stages. But the overall trend is that the larger the loading force is, the deeper the wedge is embedded, which is the result of the slip line field theory. Compared with the wedge with different tooth angles, the smaller the tooth angle of the wedge is, and the smaller the external load applied on the wedge is when the wedge is embedded in the depth of 1 mm. This shows that the sharper the wedge is, the easier it is to be embedded into the inner part of the specimen. This is consistent with the experimental data in Table 2.

In order to obtain the stress distribution in the wedge, the wedge is defined as an elastic body with large elastic modulus in the simulation environment. Repeat the above simulation steps, and get the stress distribution in the cross section of the wedge as shown in Figure 14 [19].

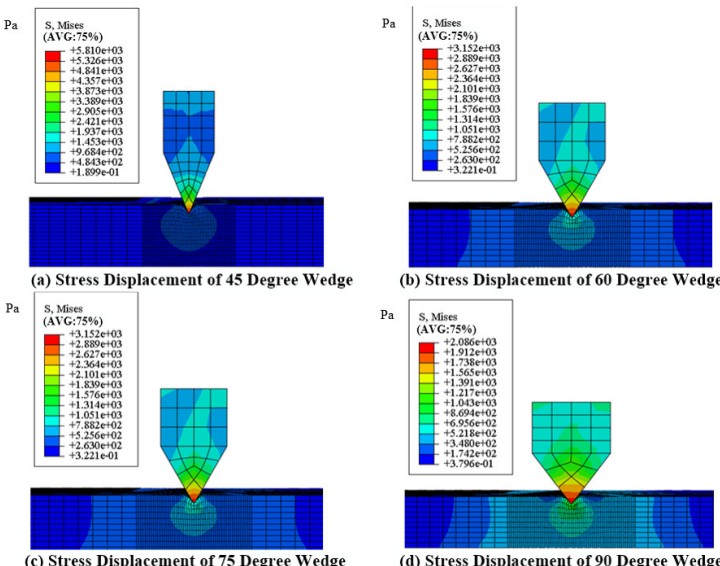

**Figure 14.** Stress Distribution in Section of Four Kinds of Tooth Angle Non-rigid Wedge after Embedding with 45°, 60°, 75° and 90° Tooth Angle.

Although the real material parameters of the wedge teeth and the marine foundation pile are not defined in the simulation, the purpose of this part of the work is to observe the stress distribution inside the wedge teeth.

Although the simulation method of defining the wedge as an elastic body with larger elastic modulus cannot completely simulate the plastic deformation of pile specimens in the process of wedge embedding, it can be seen from Figure 14 that the yield stress of the specimens is reached and the plastic deformation occurs, and the distribution of stress field is similar to that of the rigid model simulation results [20]. In order to obtain the stress distribution in the wedge more clearly, the axis of the wedge section is taken as the change path, and the stress changes of four angle wedges on the path are extracted from the simulation results. Figure 15 is obtained.

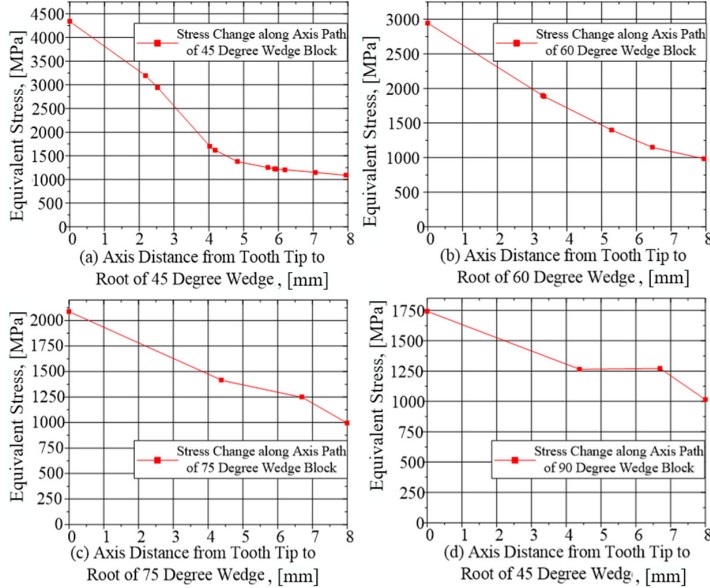

**Figure 15.** Stress Variation of Four Non-rigid Wedges along the Axis Path from Tooth Tip to Tooth Root with 45°, 60°, 75° and 90° Tooth Angle.

According to Figures 14 and 15, it can be seen that the stress concentration point is at the tip of the wedge. The stress gradually expands and decreases from the tip to the root direction. The smaller the tooth angle is, the more obvious the stress concentration phenomenon is. With the increase of the tooth angle, the stress concentration phenomenon is alleviated. In addition, considering the conclusion of the combined Equation (4), the tooth shape angle of the wedge teeth is preliminarily selected to be 75 degrees.

### 4. Design on Overall Scheme of Internal and External Clamping Variable Diameter for the Marine Foundation Pile Spreader

The clamping mechanism is the core structure of the foundation pile spreader. The above theory, experiment and simulation analysis have laid a theoretical foundation for its design. Through the previous analysis, we improve the carrying capacity and service life of the spreader. The clamping mechanism adopts the wedge tooth clamping method, and the tooth shape angle of the wedge tooth is selected to be 75 °. The structure of the hanger described in this paper is shown in Figure 16. In the Figure 16, the part I is the Large-end Clamping Component, the part II is the Motion Connector, the part III is the Hydraulic Cylinder Components with Double Out Rods, the part IV is Suspension Beam Frame Components, the part V is the -Small-end Clamping Component.

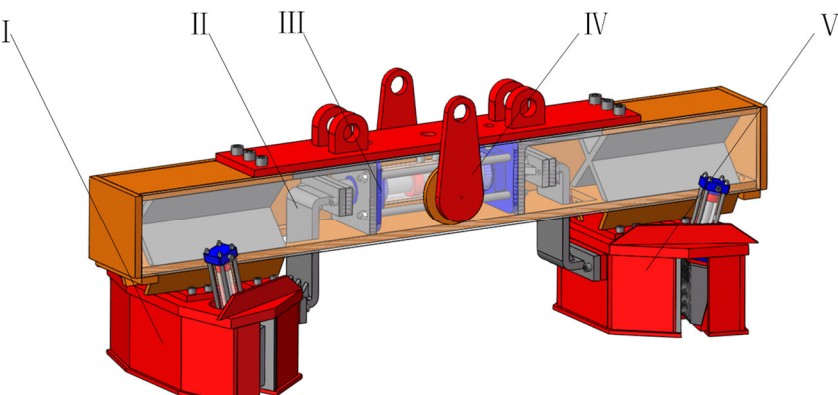

**Figure 16.** Overall Structure of Internal and External Clamping Type Marine Pile Spreader with Changeable Diameter.

The pressing teeth in our clamping mechanism need to be embedded in the base metal, and plastic deformation will occur, which is to increase the bearing capacity. However, this deformation is local deformation and will not affect the overall strength of the whole pile. On the other hand, our clamping mechanism is double-sided clamping, double-sided clamping at the same time, which is the relationship between the acting force and the reaction force. In this way, the overall damage of the foundation pile will not be affected.

The hanger is mainly composed of suspension beam frame, double-rod hydraulic cylinder, moving connector, large-end clamping component, and small-end clamping component and so on. The whole structure is mainly welded connection. For some parts, bolt connection is used to facilitate disassembly and assembly.

The suspension beam frame IV provides installation position and reliable support for other parts, and the crane completes lifting operation by welding the lifting lugs on it. The large end clamping part I and the small end clamping part V are the main working parts of the lifting tool to complete the clamping action on the pipe wall of the foundation pile. The double-rod hydraulic cylinder part III cooperates with the moving connector part II and the frame IV to complete the variable diameter action of the clamping parts on both sides. Its workflow is shown in Figure 17a–d.

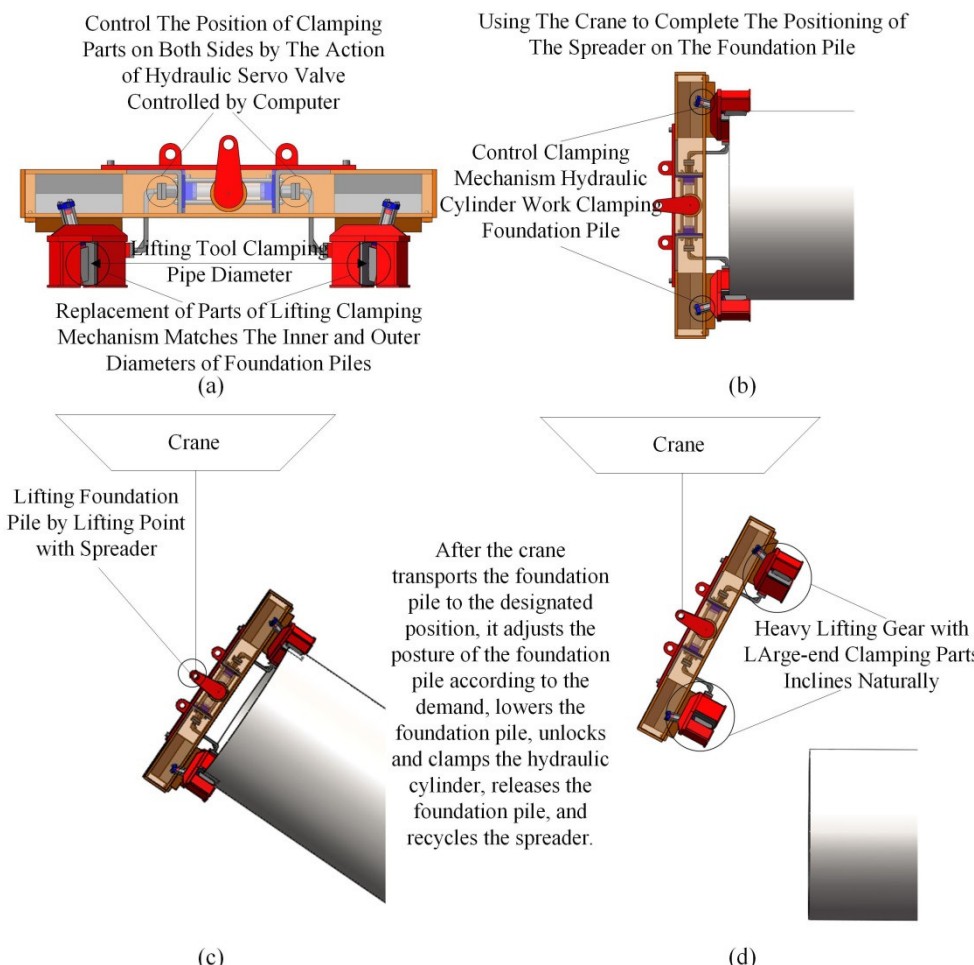

**Figure 17.** Work Flow of Lifting Foundation Pile by Using the Clamping Type Marine Pile Spreader with Changeable Diameter. (**a**) the lifting device, (**b**) the hanger is installed on the upper part of the pile, (**c**) the hoist lifts the cable, (**d**) the crane lifts, retrieves the lifting.

Firstly, as shown in Figure 17a, according to the inner diameter of the foundation pile to be lifted, the lifting device is matched with the clamped foundation pile by using the variable diameter function of the lifting device.

Secondly, as shown in Figure 17b, the hanger is installed on the upper part of the pile to be lifted by the crane. Through the control system, the hydraulic cylinder of the clamping mechanism in the clamping parts of the large end and the small end is controlled to complete the clamping of the pipe wall of the pile.

Thirdly, as shown in Figure 17c, after the hanger clamps the foundation pile, the hoist lifts the cable, and the hanger lifts the foundation pile. Through the movement of the hoist itself, the foundation pile is moved to the designated position.

Finally, as shown in Figure 17d, the crane adjusts the pile posture and slowly lowers the pile according to the actual needs (handling or piling). After lowering, the control system controls the hydraulic cylinder piston rod in the clamping mechanism to retrieve and release the foundation pile. Subsequently, the crane lifts, retrieves the lifting gear, and completes the operation process.

## 5. Conclusions

Aiming at the problem that the existing foundation pile spreaders generally have poor adaptability to the diameter of the foundation pile, the structure of the foundation pile spreader is optimized and a variable diameter form of foundation pile spreader is designed. The two clamping methods of friction clamping and wedge tooth clamping of the spreader

clamping mechanism were studied in depth. Through experimental research, it is found that under the same clamping force, the bearing capacity of the wedge tooth clamping method is three times that of the friction clamping method. And through theoretical analysis and simulation analysis, the stress of the wedge teeth at different angles during the clamping process was studied. The results show that under the same clamping force, the smaller the tooth profile angle of the wedge teeth, the deeper the depth of the wedge teeth embedded in the pile test specimen. However, due to the stress concentration point between the teeth of the wedge block, the smaller the tooth profile angle, the more obvious the stress concentration phenomenon, and the easier the tooth tip to break. Therefore, when selecting the clamping method of the wedge teeth, the selection of the tooth profile angle cannot be too large or too small. Through research in this paper, it is recommended to select a tooth angle of 75 degrees to achieve a better clamping effect.

**Author Contributions:** The first author, Z.W., conceived the framework of the article and wrote the article; the second author, Z.L., analyzed the experimental method on the clamping mechanism of spreaders; the third author, T.W., studied on the bearing mechanism of friction of lifting tools; B.Z., studied on overall scheme of internal and external clamping variable diameter for the marine foundation pile spreader. All authors have read and agreed to the published version of the manuscript.

**Funding:** This paper was funded by NSFC (Contract name: Research on ultimate bearing capacity and parametric design for the grouted clamps strengthening the partially damaged structure of jacket pipes). (Grant number: 51879063) and (Contract name: Research on analysis and experiments of gripping and bearing mechanism for large-scale holding and lifting tools on ocean foundation piles), (Grant number: 51479043).

**Institutional Review Board Statement:** The study was conducted according to the guidelines of the NSFC, and approved by the Harbin Engineering University (protocol code: 51879063 and date of: 1 January 2019).

**Informed Consent Statement:** Informed consent was obtained from all subjects involved in the study.

**Data Availability Statement:** We don't need to upload data. Readers can contact authors if they have questions. These authors can explain the data analysis.

**Conflicts of Interest:** The authors declare no conflict of interest.

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
