# Peer review of "Study on Clamping Mechanism of Internal and External Variable Diameter Lifting Tool for Offshore Foundation Pile"

_machines, doi:10.3390/machines9010019_

Round 1

Reviewer 1 Report

  1. The description of figure 11 is not consistent with the description of the axle.
  2. The correct force unit is kN, not KN.

Author Response

I am very grateful to your comments for the manuscript. According you’re your advice, we amended the relevant part in manuscript. Some of your questions were answered below.

To Reviewer #1:

Specific issues

  • Opinion 1: The description of figure 11 is not consistent with the description of the axle.

Responses to 1: We apologize for our bad chart description, we have revised the chart description and tried our best to make it easier for readers to understand, as shown in line 373-379 of Page 12. The previous content is: “From Figure 11, the wedge insertion depth t increases linearly with the increase of positive external load pressure q, while the smaller the wedge tooth angle beta is, that is, the sharper the wedge is, the deeper the wedge is.” We modify it to: From Figure 11, the wedge embedding depth t increases linearly with the increase of positive pressure q of external load, while the smaller the wedge tooth angle b and the sharper the wedge, the greater its embedding depth. In general, it is necessary to apply a large force load to the rigid wedge, so that the wedge can move down by a small amount of displacement. However, because the output of the hydraulic cylinder is constant, the tooth profile angle of the wedge determines the depth of insertion of the wedge. The key factors for the reasonable selection of the wedge tooth profile angle are of great significance in practical engineering.

  1. Opinion 2: The correct force unit is kN, not KN.

Responses to 2: As shown in the Table 4, in line 282, we modified to the correct unit notation.

************************************************

We would like to express our great appreciation to you and reviewers for comments on our paper. Looking forward to hearing from you.

Thank you and best regards.

Yours sincerely,

Zhuo Wang

Zhuo Wang

Eail: wangzhuo_heu@hrbeu.edu.cn

Jan.10.2021

Reviewer 2 Report

An elastic-dissipative mechanical system whose mass-inertial elements are in contact under a force closure is considered. Such connections are generally unileteral. It is known that unilateral ties have a number of features depending on the external vibration loading. Depending on the vibration loading, such connections can weaken and even open. Possible, It may have made sense to evaluate the critical conditions of the contract depending on the vibration load(which is characterized by the amplitude and frequency of oscillation) on the Mass-inertial elements of the system.

Author Response

I am very grateful to your comments for the manuscript. According you’re your advice, we amended the relevant part in manuscript. Some of your questions were answered below.

To Reviewer #2:

Specific issues

Thank you for your comments on this paper and we look forward to your next comments!

Responses to: Now we have finished revising the article, adding sentences and red style notes.

************************************************

We would like to express our great appreciation to you and reviewers for comments on our paper. Looking forward to hearing from you.

Thank you and best regards.

Yours sincerely,

Zhuo Wang

Zhuo Wang

Eail: wangzhuo_heu@hrbeu.edu.cn

Jan.10.2021

Reviewer 3 Report

Correct abstract, keywords.

Interesting introduction with a correct literature review.

In my opinion, the paper lacks nomenclature. All signs, abbreviations and symbols should be found here. This should come right after the abstract. Unfortunately, not all values in the paper are carefully explained and discussed by the authors.

Table 1 - what is "Horizontal Loading Force(T)" - what does "T" stand for and what is "Maximum Longitudinal Loading Force(T)" and again what does "T" stand for? I suppose it might be "A ton of strength - former T" - but that's what the authors of the paper are expected to do. Anyway, there is no space between the word description and the sign in parentheses. Should the authors not introduce some designations and give a specific unit of these forces? It is not clear. Two forces cannot be labeled the same - please change it.

Let us try to use the commonly accepted units in the manuscript and given in the SI system, so that this does not raise any doubts - kg, m, s and their derivatives.

Formula (1) - no indication of what is "m with subscript max" - please complete it.

There is no space between the text of the manuscript and the formulas, tables and figures in the paper. Please correct it.

Figure 2 is not a clear description of the axis for me - why the sign "/" between the word description and its designation, if the authors want to provide a verbal description, please separate it from the symbolic designation with a comma "," and please specify the physical unit of the quantity in square brackets "[]", eg for force "[kN]". This is true of all graphs at paper where the authors are inconsistent. Once after the "/" sign they give the sign to the given word description, and once the units in a strange way.

Table 2 - to be changed - no spaces between the word description and the unit given in brackets. For example, there is "Lateral load(T)" and it should be "Lateral load, [T]" - I tend to put units in square brackets. What sizes are in the table cells in rows 2 through 5 and columns 2 through 7? There is no clarity here and the reader may have difficulty understanding it.

Table 3 is also not very understandable - difficult to read and interpret, there is no information about physical units. The measure of deformation may be difficult to interpret. Please explain that.

Figure 4 should be amended with the guidelines given above.

I don't understand equation (3) - is it spelled correctly? Similarly, equation (2). Both equations seem to be detached from the text and are not consistent with it - the text of the paper should refer to the given equations in the paper - the authors do not do it, or they do it with too little emphasis.

Figure 11 - a completely incomprehensible description of the axis of this graph. The authors totally do not skilfully describe the charts in their manuscript. It is also incomprehensible to mention the "external pressure" unit as "KN/m" - in my opinion it should be "kN/m^2". The unit should be given in square brackets.

In my opinion, symbols in the text should be italicized - and the names of Greek letters should not be written in words - eg the word "beta" in line 323 of the work - page 11 of the manuscript.

Section 3.4 needs to be thoroughly improved. The authors say that they used the Abaqus program. They used 8 nodal spatial elements. They do not state how many nodes there were in the model, how many finite elements were there, how they assessed the convergence of the FEM model. They should present drawings presenting the finite element mesh, they should give the size of the finite element, show the discretization of the physical model into the FEM model - this is missing in the paper. Please complete this. They should clearly define the loads and boundary conditions - they should be shown in the FEM model drawing. Since we are dealing with large deformations, why do the authors decide to use 8-node linear finite elements? Did the authors check the FEM model with 20 or 27 nodal elements - such finite elements are generally used in the calculation of limit states. There is no guarantee that the 8-node model used by the authors will converge in the FEM model. Personally, I rarely use 8-node spatial elements - not for research work - only in teaching students using FEM programs with student restrictions. The model of the material used in the simulation is not clear to me. How is it possible that the yield point corresponds to zero deformation - Table 7. The deformation corresponding to the yield point is the quotient of the yield point and Young's modulus. After all, what material was used by the authors in the simulation. Please include in the engineering thesis the tensile curve of the material in question and plot the actual tensile curve on the graph. What material model do the authors assume in the FEM calculations? Do they assume large deformations and large displacements at paper - if so, how do they ensure the convergence of solutions in the FEM model? Where do they estimate the physical quantities - in nodes or points of numerical integration (how many of these points are there per 1 finite element)? Please write what the material of the variable angle mandrel was. Please give its material constants and tensile diagrams - the same as for the material of the specimens. In my opinion, the discretization of the tool is too poor - it should be more accurate. Did the authors solve the contact issue here?

Figure 12 does not specify the units in which the von Mises stresses are given, and appears to contain "Chinese" stamps. Please complete and correct it. What does "75%" mean in this figure? The presented graphs are almost similar - it is worth supplementing them with magnifications and line graphs along the thickness or in the direction of the tool's operation. In the plots of isolines - enlarged, it is worth giving the axes with physical coordinates.

Figure 13 illegible in axle designation - merging. It is worth comparing the curves for several tool geometries on one graph - please add another two drawings to the paper. Please pay attention to the units. What is "Fritting line" - please explain.

Figure 14 - See notes to Figure 12.

Figure 15 - See the notes to Figure 13.

The paper is interesting, but in my opinion it is suitable for publication. I recommend "major revision". I am waiting for a revised version of the paper.

Author Response

(The authors gave the same response as above.)

Round 2

Reviewer 3 Report

The all corrections suggested for the previous version of the paper have been included by the Authors in the new thoroughly revised version of the paper. The paper put in by the Authors to improve the article meant that the paper has improved quality, is more readable and brings new elements. I recommend paper for publication in the Machines Journal.